# Co-Combination of Pregabalin and *Withania coagulans*-Extract-Loaded Topical Gel Alleviates Allodynia and Hyperalgesia in the Chronic Sciatic Nerve Constriction Injury for Neuropathic Pain in Animal Model

**DOI:** 10.3390/molecules27144433

**Published:** 2022-07-11

**Authors:** Anam Asghar, Muhammad Naeem Aamir, Fatima Akbar Sheikh, Naveed Ahmad, Mervat A. Elsherif, Syed Nasir Abbas Bukhari

**Affiliations:** 1Department of Pharmaceutics, Faculty of Pharmaceutical Sciences, Government College University Faisalabad, Punjab 38000, Pakistan; dr.anumasghar@yahoo.com; 2Department of Pharmaceutics, Faculty of Pharmacy, The Islamia University of Bahawalpur, Bahawalpur 63100, Pakistan; 3School of Pharmacy, Newcastle University, Newcastle Upon Tyne NE1 7RU, UK; 4Al-Raziq College of Pharmacy, Sargodha 40100, Pakistan; fatimatahir303@gmail.com; 5Department of Pharmaceutics, College of Pharmacy, Jouf University, Sakaka 72388, Al Jouf, Saudi Arabia; nakahmad@ju.edu.sa; 6Chemistry Department, College of Science, Jouf University, Sakaka 72388, Al Jouf, Saudi Arabia; maelsherif@ju.edu.sa; 7Department of Pharmaceutical Chemistry, College of Pharmacy, Jouf University, Sakaka 72388, Al Jouf, Saudi Arabia; sbukhari@ju.edu.sa

**Keywords:** co-combination gel, pregabalin, *Withania coagulans* extract, chronic constriction injury, topical delivery

## Abstract

The current study reports the fabrication of co-combination gel using Pregabalin and *Withania coagulans* fruit extract to validate its effectiveness for neuropathic pain in chronic constriction injury (CCI) rat models. Three topical gels were prepared using Carbopol 934 through a pseudo-ternary phase diagram incorporating the Pregabalin (2.5%), *Withania coagulans* extract (2%), and co-combination of both Pregabalin (2.5%) and *Withania coagulans* extract (2%). Gels were characterized. FTIR showed a successful polymeric network of the gel without any interaction. The drug distribution at the molecular level was confirmed by XRD. The AFM images topographically indicated the rough surface of gels with a size range from 0.25 to 330 nm. DSC showed the disappearance of sharp peaks of the drug and extract, showing successful incorporation into the polymeric network of gels. The in vitro drug release of co-combination gel was 73% over 48 h. The mechanism of drug release by combination gel was Higuchi+ fickian with values of *n* (0.282) and R^2^ (0.947). An in vivo study for pain assessment via four methods: (i) heat hyperalgesia, (ii) cold allodynia, (iii) mechano-hyperalgesia, and (iv) dynamic mechano-allodynia, confirmed that topical treatment with co-combination gel reduced the pain significantly as indicated by the *p* value: R1 (*p* < 0.001), R2 (*p* < 0.001), R3 (*p* < 0.015), and R4 (*p* < 0.0344). The significance order was R2 (****) > R1 (***) > R3 (**) > R4 (*) > R5 (ns).

## 1. Introduction

Neuropathic pain is caused by lesions, tissue damage, and disease of the somatosensory system including peripheral fibers and central neurons [1]. The capacity to experience pain involves coordinated reflexes and behavioral responses. The pain resulting from damage to the nervous system (peripheral neuron, dorsal root ganglion, and central nervous system) is said to be neuropathic pain [2]. Its causes also include diabetes, trauma, channelopathies, autoimmune diseases, and infection [3]. Neuropathic pain is characterized by allodynia and hyperalgesias either simultaneously or independently. The National Institute for Health and Care Excellence (NICE) guideline recommends the use of Amitriptyline, Pregabalin, Duloxetine, and Gabapentin for the treatment of neuropathic pain [4]. However, the use of these drugs is limited due the side-effects associated with them.

The precise mechanism of neuropathic pain has not been clear. However, previous studies have indicated that inflammation plays a major role in initiating and maintaining neuropathic pain. Immune-like glial cells in the injured nerves and spinal cord are responsible for inflammatory mediator release and also the release of cytokines IL-6, IL-1β, and TNF-α, which trigger and maintain hyperalgesias in the CCI injury sciatic nerve model [5]. Pregabalin was used in Europe in July 2004 for the treatment of neuropathic pain and as an adjuvant in the partial treatment of epilepsy. In December 2004, the FDA approved Pregabalin for the treatment of neuroglia, diabetic neuropathy, and post-herpetic neuroglia [6]. Pregabalin reduces neuronal excitability in the central nervous system by reversibly binding to the alpha 2-delta subunit of the calcium channel [7] and reduces the release of synaptic neurotransmitters. Oral Pregabalin is associated with a number of side-effects such as weight gain, somnolence, dizziness, and peripheral edema [8]. 

Delaying treatment of the peripheral area may worsen the pain, and topical agents serve as the first-line agent in overcoming the pain with fewer side-effects in comparison to the oral agents. Topical agents reduce the pain by acting on sensory nerve endings and adjacent cellular elements. Only a few topical preparations are licensed and available for neuropathic pain, such as capsaicin cream [9]. 

Plants are a natural source of maintaining good health and recovering from a disease with fewer side-effects [10]. According to the research carried out by the World Health Organization (WHO), more than 80% of the world population depends upon plants as healers [11]. *Withania coagulans*, a natural plant, is known as Paneer Doda and belongs to the family Solanaceae located in Pakistan and its surroundings. *Withania coagulans* is usually known as a cheese maker in the Indian language and paneer in Pakistan because its fruit and leaves have milk-coagulating properties. *Withania coagulans* has been used in a number of diseases and disorders due to the presence of Withanolides [12]. Active constituents are *Withaferin* A, *Withanolide*, and *Coagulins* [13]. It is used in diabetes-induced neuro and nephropathy [14]; it has anti-nociceptive, antimicrobial, antifungal, anti-inflammatory [15], immunosuppressant, insomnia, and antidepressant properties; it is used in tooth pain, asthma, withdrawal symptoms of morphine dependence [16], hepatoprotective, and breast cancer.

Previously, Pregabalin and the *Withania coagulans* dosage form were prepared separately or in combination with other drugs for oral/topical use. However, co-combination gel of the extract and Pregabalin has not yet been evaluated [17]. 

The purpose of this study was to evaluate the individual effect of Pregabalin and *Withania coagulans* gel alone and as co-combination gel using both for the treatment of neuropathic pain in chronic constriction injury in a rat model. The prepared topical gels were applied twice a day for four days. The animal were observed for behavioral responses including (i) heat hyperalgesia, (ii) cold allodynia, (iii) mechano-hyperalgesia, (iv) dynamic mechano-allodynia, and (v) depression-like behavior. Responses were noted ans statistical analysis was done.

## 2. Materials and Methods

### 2.1. Materials

*Withania coagulans* fruit was purchased from the local market. Pregabalin and Carbopol 934 were generously gifted by Brookes Pharma (Pvt.) Ltd., Karachi and Saffron Pharmaceuticals (Pvt.) Ltd., Faisalabad, Pakistan, respectively. Frankincense oil was purchased from Co Natural™. Transcutol^®^ P and Tween 80 were acquired from Gattefosse, France (Nanterre, France) and Sigma-Aldrich (St. Louis, MO, USA), respectively. Xylazine and Ketamine were used to anaesthetize the animals. All other chemicals and reagents used were of analytical grade. Distilled water was used during the whole study.

### 2.2. Preparation of Pregabalin and Withania coagulans Topical Gel

#### 2.2.1. Obtaining *Withania coagulans* Extract

The procedure for obtaining *Withania coagulans* fruit extract has been described in [18]. Briefly, *Withania coagulans* fruit was obtained from the local market and was identified under accession no.15063 Hazara University Herbarium Pakistan. The fruits were soaked in a mixture of chloroform and methanol (1:1) for 4–5 days. After that, the mixture was passed through a muslin cloth and the extract was collected followed by drying on a rotary evaporator to obtain the viscous paste. 

#### 2.2.2. Preparation of Micro-Emulsion and Topical Gel

The micro-emulsion was prepared as reported [18]. Briefly, the micro-emulsion was prepared by using a surfactant mixture (Tween 80, co-surfactant Transcutol P), and Frankincense oil was prepared by a pseudo-ternary phase diagram. A surfactant/co-surfactant mixture was added in the Frankincense oil and water was added dropwise gradually until a clear micro-emulsion region appeared. The prepared micro-emulsion was incorporated in Carbopol 934 hydrogel (1%). After that, the Pregabalin (2.5%), *Withania coagulans* extract (2%), and the combination of both Pregabalin and *Withania coagulans* were mixed in the Carbopol hydrogel to prepare three different formulations. The prepared gel was applied topically twice a day at 8 am and 12 pm on CCI rat models. 

### 2.3. Characterization of Topical Gels

The drug–excipient interaction of the ingredients of the formulation was assessed through FTIR and DSC. The prepared gel was characterized through AFM and XRD.

#### 2.3.1. Drug–Excipient Compatibility Studies

The physical and chemical interactions between the drug and excipients were determined by the FTIR technique. A 100 mg sample was mixed thoroughly with potassium bromide and was compacted under a vacuum pressure of 12 psi for 3 min. The disc was mounted in a suitable holder in a Perkin Elmer IR spectrophotometer. The spectra were recorded within 2000 cm^−1^–400 cm^−1^. The resultant spectra of the pure drug obtained were then compared with the spectra of its physical mixture.

#### 2.3.2. DSC and TG Analysis

The thermal and phase transition behavior changes of the drug, extract, and prepared gels were determined by the DSC technique with nitrogen purging. The heating rate of 20 °C/min was used over a temperature range of 50–250 °C. A slander aluminum sample pan was used for placing the sample.

#### 2.3.3. X-ray Diffraction of Gels

The samples were placed on a special plane glass. The small-angle X-ray diffraction was observed by using a super-speed diffractometer with Ni, a filter, and Cu radiation. The voltage of the tube was set to 25–45 kV and the tube current was maintained at 100–200 mA. The samples were scanned from 2° to 70°, 2θ [19].

#### 2.3.4. Atomic Force Microscopy

Atomic force microscopy (AFM) is an advanced technique for surface topographic imaging. It employs height, diameter, and other surface properties such as roughness, surface area, maximum depth, volume, and energy as compared to transmission electron microscopy (TEM) and scanning electron microscopy (SEM). The surface morphology of topical gels was observed using AFM (Park systems Corp Model: XE-7) operating in intermittent–contact AFM mode in air and a silicon cantilever tube.

### 2.4. Ex Vivo Skin Permeability Studies

Ex vivo skin permeation studies were performed for topical gels using the excised skin of rats with an average weight of 200–300 g. Approval for animal studies was taken from Ethical Committee Government College University, Faisalabad (Ref no: GCUF/ERC/Pharm 717/2021). Rats were sacrificed by excess chloroform, and hairs from the abdominal skin were removed. The skin was removed from the upper abdominal area by using forceps. The skin was rinsed with phosphate buffer pH 6.8 and then skin permeation studies were conducted using a Franz diffusion cell. It consists of a donor and a recipient chamber. The donor compartment was opened and exposed to the atmosphere. The skin with an area of almost 1.76 cm^2^ was mounted between the compartments with the stratum corneum facing the donor compartment. The receptor compartment was filled with pH 6.8 phosphate-buffered solution. The temperature of the media (pH 6.8) was maintained at 37 ± 0.5 °C and the rotation speed was kept 100 rpm. Gel (1.0 g) was applied on the skin fitted on the Franz diffusion cell. Samples were withdrawn from the receptor compartment at predetermined study intervals and replaced with fresh buffer after each time [20]. The amount of permeated gel was determined by a UV/Spectrophotometer at 405 nm for Pregabalin and 340 nm for the extract [21]. The amount of drug and extract permeated was calculated by plotting a graph against time. Permeation flux was calculated using the equation given below.
(1)$$j=\frac{\mathrm{dQ}}{\mathrm{dt}\times A}$$
where J = Jess is the steady-state permeation flux (µg/cm^2^/h), dQ/dt = amount of the drug passing through the skin per unit time, and A = area of skin tissue (cm^2^).

### 2.5. In Vitro Drug Release Studies of Topical Gels

In vitro drug release studies were carried out by the dialysis bag method. One gram of the gel (Pregabalin-loaded, *Withania coagulans* extract-loaded, and combination of both) was placed separately inside the dialysis bag. The dialysis bag was tied at both ends and then placed in a 100 mL beaker filled with phosphate-buffered solution (pH 6.8). The beaker was then placed in a shaking water bath at a constant temperature of 37 ± 0.5 °C for 24 h. The sample of 0.5 mL was drawn from the recipient compartment and replaced by 0.5 mL of fresh pH 6.8 phosphate buffer solution. The samples were withdrawn at intervals of (5, 15, 30, 60, 90, 120, 150, 180, 210, 240, 300, 360, 420, 480, and 1440 min) [20]. To each Pregabalin sample, ninhydrin solution (1%) prepared in DMF was added [22,23]. After that, this mixture was heated in a water bath at 70 °C for 15–20 min. Complexation took place as indicated by the change in color of ninhydrin from pale yellow to purple. Then, the absorbance was taken at 405 nm for Pregabalin. The same procedure was repeated for extract-loaded gels without adding ninhydrin solution. The absorbance was taken at 340 nm for extract-loaded gels. The percent release was calculated and expressed in graphical form.

### 2.6. In Vivo Studies

#### 2.6.1. Animals

Male albino rats weighing 170–250 g and maintained in a 12 h light/dark cycle were used in the experiment. Animals were allowed free access to food and water. All animal procedures were performed in compliance with the UK Animals (Scientific procedures) Act 1986, according to the rules set forth by the Ethical Committee of Department of Pharmacy, Government College University, Faisalabad. Approval of the study was granted with the reference no: Pharm/21/717 on 2 October 2021.

#### 2.6.2. Induction of Neuropathic Pain Model

Animals were anesthetized by Xylazine and Ketamine intraperitoneally. The sciatic nerve was ligated as explained previously by [24] and is shown in Figure 1. Animals were placed in the prone position on a cotton pad during the procedure. The animals’ left thigh was elevated and the fur from the posterior surface was removed with the help of an electric clipper. The exposed skin was cleaned with povidone iodine 10% *w*/*v* solution (Pyodine^®^, Brookes Pharma, Karachi, Pakistan). After the surgical procedure, the sciatic nerve was exposed at the middle of the thigh by blunt dissection (approximately 4–5 mm deep) through the biceps femoris. Proximal to the sciatic trifurcation, 10 mm of the nerve was set free from the tissue with a micro scissor. Two ligatures (Chromic catgut suture 4/0 metric 2; Ethicon, Karachi, Pakistan) were tied with a double knot 1 mm apart for trifurcation of the sciatic nerve. The constriction of the nerve was minimal and immediately stopped on observing a brief twitch. The muscle layer was closed by a silk-braided suture 2/0 metric 2. Animals were then treated with the developed formulation and responses were noted.

#### 2.6.3. Animal Treatment Groups

Animals were treated by applying topical Pregabalin gel 2.5%, *Withania coagulans* extract gel 2%, and the combination of both Pregabalin and *Withania coagulans*. The control group received the Pregabalin drug solution in the same concentration as that given to the treated-group animals. Animals were assigned into the following groups. A uniform quantity of all gels was applied to the hind paw twice a day at 8 am and 12 pm.

Rats were divided into five groups (*n* = 6 each):R1 = Topical application with *Withania coagulans* extract-loaded gel;R2 = Topical application of Pregabalin + *Withania coagulans* gel combination;R3 = Topical application with Pregabalin-loaded gel;R4 = Topical application with Pregabalin drug solution;R5 = No Treatment group.

### 2.7. Pain Assessment

Animals were transferred to an elevated wire mesh bottom table and were acclimatized for 40–45 min before assessment. Heat hyperalgesia, cold allodynia, mechano-hyperalgesia, and dynamic mechano-allodynia were assessed before surgery and on day 0, 1, 2, 3, and 4 after surgery. Behavioral responses were measured 30 min after applying the gel to the hind paw.

#### 2.7.1. Evaluation of Thermal Hyperalgesia

##### Hot Plate Method

The mid-planter surface of the operated left hind paw was touched to the heated surface (hot plate) maintained at 56 °C [4,25]. The heat source was adjusted at the beginning of the experiment to yield a paw flick within 10 s. The paw withdrawal latency (PWL) and duration were recorded with a minimal time of 5 s and maximum of 15 s. Licking and lifting was considered as the Paw Withdrawal Response.

##### Water Bath Method

The mid-planter surface of the operated left hind paw was immersed in a hot water bath maintained at 50 °C. PWL and duration were recorded with a minimum of 5 s and maximum of 10 s [26].

#### 2.7.2. Evaluation of Cold Allodynia

##### Cold Water Method

In this test, ice-cold water was taken and the mid-planter surface of the operated left hind paw was immersed in ice-cold water. The paw withdrawal duration was noted. The minimum duration was 5 s and the maximum was 15 s [27].

##### Acetone Spray Method

A 50 µL volume of acetone was sprayed delicately on the mid-planter surface of the left hind paw of the operated side using a syringe without touching the skin of animals. PWL was calculated with a minimum of 5 s and maximum of 15 s [28].

#### 2.7.3. Mechano-Hyperalgesia

##### Pin Prick Test

Animals were placed on an elevated grid, and the tip of the ordinary needle (safety pin) was pressed against the mid-planter surface of the operated left hind paw of the animals. The paw withdrawal duration was noted as minimum for 5 s and maximum for 15 s [4,29].

#### 2.7.4. Dynamic Mechano-Allodynia

##### Cotton Bud Test

In this test, the mid-planter surface of the left hind paw s of the animal was stroked by a cotton bud. Lifting, moving forward, and licking was considered as the paw withdrawal response. The cut-off time was maintained at 15 s [4,30].

##### Paint Brush Method

The response of animals toward a smooth paint brush is considered as allodynia because rats rarely withdraw from this stimulus. Rats were placed in a wire mesh cage and a stimulus from the heels to the toes of the rats was applied by a smooth paint brush for 15 s. A paw withdrawal reaction within 15 s was considered as dynamic mechanical allodynia [31].

#### 2.7.5. Evaluation of Depressive Behavior

##### Tail Suspension Method

In this test, each mouse was suspended via the tail on a horizontal bar (50 cm from the floor) using adhesive tape. Tape was attached 4 cm above from the tip of the tail. The duration of immobility of the tail was recorded from 5 s to 6 min. The immobility time was considered as the absence of any escape-oriented behavior [32].

### 2.8. Statistical Analysis

Animals were assigned in five treatment groups, each having 6 rats. Two-way analysis of variance was applied (Days X Treatment) followed by a *post hoc* Bonferroni test. Results are presented as mean ± SD. A value of *p* < 0.05 was accepted as significant.

## 3. Results and Discussion

### 3.1. Characterization of Topical Gels

#### 3.1.1. Drug–Excipient Compatibility Studies

The compatibility between drug and excipients was determined by the FTIR technique (Table 1 and Figure 2).

The peaks reported in the literature for *Withania coagulans* extract occur at 1733 cm^−1^, 1446 cm^−1^, 1382 cm^−1^, 1162 cm^−1^, and 936 cm^−1^. The FTIR peaks of our sample of *Withania coagulans* appear at 3295 cm^−1^, 2944 cm^−1^, 1647 cm^−1^, 1401 cm^−1^, and 1013 cm^−1^. The peaks between 3000 and 2800 cm^−1^ are due to lipids and are responsible for CH stretching vibrations. The band at 2944 cm^−1^ is due to CH_2_ and CH_3_ groups, so it can be related to the alkenes group present in *withania coagulans* extract. The peak at 1647 cm^−1^ is due to the stretching vibration of C=C, resulting from the deformation of the aromatic ring of flavonoids [34]. The peak at 1013 cm^−1^ indicates C-O structure vibrations of β-isomers of hydroxyl ketones [35].

The Frankincense oil contains diterpene alcohol. The strong peak at 1162 cm^−1^ is due to the presence of lipids and alcohol (stretching of C-O bond and bending C-OH group of alcohol). The two bands in alcohol appearing at the 1300–1450 cm^−1^ position are due to the bending vibration of CH_2_-CH_3_ aliphatic groups. Here, in our IR spectra of oil, this CH_2_ CH_3_ bending is observed at 1446 cm^−1^. The peak at 1733 cm^−1^ is due to the carbonyl functional group of COOH [36]. The absorption peaks for Smix appearing at 1736 cm^−1^ are due to C=O stretching and those at 1647.5 cm^−1^ are due to the amide group [37]. The characteristic peaks of Pregabalin are present in the gel formulations as well, which indicate that there was no drug–excipient interaction. The sharp peaks still remain intact in all three topical gels.

#### 3.1.2. DSC and TG Analysis

The DSC curve of the pure drug exhibits a sharp endothermic peak at 196.46 °C, resulting from its melting point (Figure 3). The DSC curve of *Withania coagulans* extract appears at 111.26 °C. The formulated topical gels show broader endothermic peaks. These peaks are closer to the melting point of the extract. The drastic shift in the endothermic peak of the Pregabalin-loaded gel from 196.4 °C to 140 °C is due to the incorporation of Pregabalin in Carbopol gel. The shift/intensity of the drug-loaded gel is lower than that of the pure drug. This might be due to the incorporation of Pregabalin into the gel, resulting in the change in t from a more crystalline to less crystalline/amorphous form/disordered form [38]. This leads to the formation of a new peak, shifting it toward lower temperature, and also gives an indication of the formation of a new linkage with the gel. This leads to the formation of a new phase due to the low degree of crystallinity due to complexation [39]. There is a slight shift in the endothermic peak of the *Withania coagulans*-loaded gel, which is still closer to the melting point of *Withania coagulans*.

The TGA analysis of the drug (Pregabalin) shows that there is no weight loss up to 205 °C [40]. At 115–111 °C, the weight loss is attributed to the desorption of surface water or decomposition of the organic contents of fruit extract of *Withania coagulans* [41]. The TGA curve for the pure drug shows that the mass remains constant with temperature but falls when it comes closer to its melting point. A similar effect is observed in all remaining gels. These findings confirm that excipients and moisture contents have no adverse effects on formulations [38]. There is a slight change in the peak from 115.26 °C to 126 °C. In case of the extract-loaded gel, the weight loss is found to be 1.05%, which may be associated with the decomposition of the organic contents of the extract. In a similar way, the weight loss in the case of the drug-loaded gel and the combination of the extract and drug-loaded gel is found to be 1.37% and 0.49%, respectively. 

#### 3.1.3. X-ray Diffraction of Gels

The X-ray diffractogram confirms the crystalline nature of Pregabalin, as indicated by the number of sharp and intense peaks (Figure 4). The XRD pattern of pure Pregabalin at 2θ shows characteristic peaks at 9.4, 19.04, 38.5, 40, and 49.9 [42].

The X-ray diffractometer of the extract and all three gels confirms the amorphous nature due to the absence of sharp peaks [17,19]. It is clear that Pregabalin characteristic peaks are modified in the same thermal events, suggesting a partial amorphous nature in all the gels.

#### 3.1.4. Atomic Force Microscopy

AFM gives an idea about the 3D topographical image of ingredients with others and their actual behavior [43] It is an indirect method that supports characterization. This behavior can be observed by AFM under high resolution. The AFM topographical imaging of all three gels is depicted in Figure 5 [38]. AFM analysis of all topical gels indicates that the height of the topical gel is as follows: Pregabalin pure is (0.61 µm), Pregabalin-loaded topical gel is (0.92 µm), and Pure extract is (0.25 µm), and extract-loaded gel is (330 nm), and the combination of extract and Pregabalin gel is (140.4 nm). These results show that all gels are in the range from 0.25 µm to 330 nm. After loading into the gel, the size of the extract and Pregabalin decreases as compared to pure gels of both.

### 3.2. In Vitro Permeability Studies

Permeability studies showed that the permeation rate was higher in the case of drug-loaded gel as indicated by the flux (4.5 µg/cm^2^/h); then, co-combination gel was (4.095 µg/cm^2^/h), less than that in the case of extract-loaded gel (2.33 µg/cm^2^/h). This indicates that Pregabalin permeated within 8 h maximally, and *Withania coagulans* extract-loaded gel had a lower flux in 6–8 h, which showed maximum extract penetration in 8 h as shown in Figure 6. The permeation of co-combination gel was 4.09 µg/cm^2^/h, which was higher than the extract-loaded gel. However, the co-combination permeated slowly through the skin up to 24 h. This shows that by incorporating the combination of drug and extract into the gel, the permeation slowdown and drug is released in a sustained manner. The addition of extract into the gel along with Pregabalin resulted in a greater entanglement and, due to this, the permeation decreased. This study is supported by [44]. The decrease in Jess value indicates that a small mesh size of gel matrix or compact network in the gel may hinder the drug/extract permeation through the skin and therefore be released. A similar study was reported by [45]. Furthermore, a reduction in drug/extract diffusion across the skin gives an indication that a maximum portion of the drug is available at the local site for providing maximum topical effect.

### 3.3. In Vitro Drug Release and Mechanism of Drug Release

The release of active ingredient from the topical gel is depicted in Figure 7 and Table 2. It was noted that the drug release in the case of Pregabalin-loaded topical gel was 97% in 8 h and the drug release in the case of extract-loaded gel was 81%. By contrast, the combination of Pregabalin and extract-loaded gel showed a release of 73% in 24 h. In the case of Pregabalin-loaded gel, initially, within 15 min, 36% of the drug was released. In 6 h, 53% and, in 8 h, 97% of the drug was released, whereas, in the case of extract-loaded gel, there was an initial burst release of 65% of the drug in 15 min. Then, there was sustained release of the drug up to 8 h (80%), and then in 24 h, 81% of the drug was released. This shows extracts released from the gel in three phases. The initial burst release in the case of the extract-loaded gel may be attributed to the soft surface of the *Withania coagulans* extract, which was a smooth pasty type. The second sustained release was 80% in 8 h and controlled release in 24 h (81%). The similar results were reported by [46] for the Gingko biloba extract release profile.

The drug release order was Pregabalin-loaded gel > extract-loaded gel > drug + extract-loaded gel. The decrease in the drug release from combination gel (extract + drug) was first due to the high viscosity of the gel, having a high efficiency in entrapping the drug within the polymeric network of the gel. Secondly, the presence of the extract may interfere with the release of the drug [47]. In the case of extract + drug combination gel, 19% of the drug was released within 15 min, and then up to 6 h, 54.5% of the drug was released. After 24 h, a 73% release of the drug was observed.

The mechanism of drug release by all three gel formulations is different. The value of R^2^ in the case of the extract-based gel was 0.9528; in the case of the drug-loaded gel, the R^2^ value was 0.8628; in the case of co-combination gel, it was 0.9584. Extract-loaded gels followed the Higuchi square root law, and the mechanism of drug release was found to be diffusion-controlled as the value of *n* < 0.5, i.e., (0.124). In the case where the drug-loaded gel drug release mechanism was found to be zero order and with the further optimization by the Krosmeyer and Poppas model, it was found that the value of *n* = 0.206, which is <0.5, showing that the diffusion mechanism was involved in it. In the case of co-combination gel (E + F), the Higuchi model was involved and the value of *n* < 0.5, i.e., 0.282, showing that diffusion was involved. The drug was released by the diffusion phenomenon through the swollen gel matrix. A similar release mechanism was observed by [48]. The results for kinetic modelling are shown in Table 2 and Figure 7.

### 3.4. In Vivo Studies

#### 3.4.1. Effect of Different Pregabalin and *Withania coagulans* Combination Gels on Neuropathic Heat Hyperalgesia

The chronic constriction injury produced a decrease in nociceptive-threshold hyperalgesia, as observed by the heat stimuli response. After CCI, the animals showed an immediate response to heat stimuli, as performed by the hot plate method and by using the hot water bath. Animals withdrew the paw within 1–4 s on day 0. Conversely, treatment with topical gels and solution diminished the CCI-induced behavioral changes and responses in animal groups. After application of different treatments as mentioned earlier, the animals’ response became delayed with each passing day, and after the 4th day, there was a significant increase in response behavior of animals of all groups, as shown in Figure 8a. The duration of paw licking/withdrawal increased with the day after application of different treatments. The results were calculated by taking R5 as the control. The *p* value was found to be in R1 (*p* < 0.001), R2 (*p* < 0.001), R3 (*p* < 0.015), and R4 (*p* < 0.0344). A significant improvement was seen in the case of the combination of gel using Pregabalin and *Withania coagulans*. The significance order was R2 (****) > R1 (***) > R3 (**) > R4 (*) > R5 (ns). Similar results were reported by [4,48].

In the case of the hot water bath method, upon dipping of the paw of rats, an immediate withdrawal of the paw was seen on day 0 of CCI. After applying different treatments, it was observed that the response of the animals started increasing in seconds from day 1 to day 4. This shows that the application of gels reduces the nociceptive pain in the rats’ model. As shown in Figure 8b, the *p* value was found to be in the following order: R1 (*p*<), R2 (*p*<), R3 (*p*<), and R4 (*p* < 0.069) and significance order was found to be R1 and R2 (****), R3 (ns), and R4 (ns).

#### 3.4.2. Effect of Different Pregabalin and *Withania coagulans* Combination Gels on Neuropathic Cold Allodynia

The chronic constriction injury produced an increase in hyperalgesia, as observed by the cold stimuli response. The exposure to ice-cold water of different groups of animals produced an immediate/exaggerated withdrawal response to noxious painful stimuli and hastened the flexion to the cold stimuli. The response noted was from 0 to 8 s on day 0. This increased with each passing day. The topical application of gels resulted in the reduction in CCI-induced behavioral changes in all groups of rats. A significant delay to noxious pain stimuli exposed to cold water and acetone was observed from day 0 to day 4, as shown in Figure 9. The order of delay was in following manner: R2 (14.80 s) > R4 (12.38 s) > R3 (7.45 s) > R1 (6.33 s) > R5 (3.21 s). R5 was taken as the control. The *p* value was found to be <0.001, showing that it was significant. Multiple comparisons showed that the *p* value was in the following order: R1 (*p* < 0.0.001), R2 (*p* < 0.001), R3 (*p* < 0.001), and R4 (*p* < 0.001). These results indicate that all formulations showed a significant effect in reducing pain after the application of topical gels in comparison to R5. The significance for all formulations was noted to be (****). Keeping in mind that the delay response was more prominent in the case of co-combination gel, it showed a better efficacy in alleviating the pain behavior in rats. Similar findings were reported by [4,49,50].

#### 3.4.3. Effect of Different Pregabalin and *Withania coagulans* Combination Gels on Neuropathic Dynamic Mechano-Hyperalgesia

The rats’ groups of CCI developed significant hypersensitivity to mechano-stimuli (pin prick response) and dynamic stimuli (cotton bud and paint brush response) (Figure 10). It was observed that PWL or the moving forward duration was increased in all groups, as compared to the control group on day 0. It was from 1.53 s to 7.235 s as compared to the control group in the case of the pin prick test, and the duration of flexion withdrawal evoked by pin prick continued to increase from day 0 to day 4. It was observed that by applying topical gels to different groups, the pain threshold was decreased in the rats, and conversely, moving forward/paw withdrawal responses were increased. This trend was maintained at the end of the study. At the end, the increase in stimulus response duration was R2 > (14.805 s), R4 > (12.38), R3 > (7.45 s), R2 > (6.33 s), and R5 > (3.21 s). The *p* value was found to be significant in all the topical formulations with *p* value < 0.0001. The results are shown in Figure 10a,b below and a significance trend (****) was observed in all cases, showing that all formulations show effectiveness in comparison to the control in order to reduce pain stimuli. A similar trend was observed in the case of dynamic mechano-hyperalgesia (paint brush method). Ipsilateral paw withdrawal and the moving forward response were increased with each passing day after surgery. The pain threshold was much delayed in the case of the pain brush method, as little pain was evoked by using the paint brush. The *p* value was found to be significant in all formulations in comparison to the control, *p* < 0.0001 (****). These outcomes were consistent with the outcomes reported by [51], showing that the administration of plant extract reduces neuropathic pain in the CCl rat model. These findings are similar to those explained by [52] using plant extract *Cassia artimisiodes* for diabetes-induced neuropathic pain.

#### 3.4.4. Depressive Behavior Analysis

Depressive behavior was analyzed by the tail suspension test (Figure 11). CCIs produced depressive behavior in rat models. CCI injury led to an increase in the escape tendency in the rat model. It was noticed that there was a decrease in the escape behavior or the taking out of their tail from the cold water cylinder with the increase in days after application of topical formulations. The escape tendency was higher at day 0, and in different groups, it was different. It ranged between 3 and 17 s on day 0. As we applied topical formulations, a decrease in depressive behavior was noted with the increase in time duration for escape/tail withdrawal responses of rats. It was observed that the decrease in withdrawal response was R2 (17.25 s) > R1 (13.8 s) > R4 (12.27 s) > R3 (9.75 s) > R5 (8.9 s). The *p* value was found to be in following order: R2 (*p* < 0.0001), >R1 (*p* < 0.041), >R4 (*p* < 0.001), and R3 (ns) in comparison to control R5. The significance order was R2 (****), >R1 (***), >R4 (**), and R3 (ns). These results suggest that although all formulations show good results in comparison to the control R5 group, the co-combination of the extract and Pregabalin showed more effectiveness in lowering the depressive behavior with more delay in the response to tail withdrawal from cold water (17.25 s with *p* value < 0.0001 ****). Similar outcomes were reported by showing a delay in tail flicking response after the administration of extract [53]. 

## 4. Conclusions

Three different gel formulations (Pregabalin-loaded, *Withania coagulans* extract-loaded, and co-combination of both) were used in the current study to find their effectiveness in chronic constriction injury rat models. The gels were successfully prepared and characterized. An in vivo study was conducted by inflicting chronic constriction injury on the sciatic nerve. The neuropathic pain assessment was observed by four different methods: (i) heat hyperalgesia, (ii) cold allodynia, (iii) mechano-hyperalgesia, and (iv) dynamic mechano-allodynia. The responses of rats to these stimuli were noted and the *p* value was calculated. The *p* value of each response was compared with the control group and solution of the drug, which was R1 (*p* < 0.001), R2 (*p* < 0.001), R3 (*p* < 0.015), and R4 (*p* < 0.0344). The order of significance in heat hyperalgesia was R2 (****) > R1 (***) > R3 (**) > R4 (*) > R5 (ns). The delay in response was more prominent in the case of co-combination gel, showing better efficacy in alleviating the pain behavior in rats, as also indicated by graphical presentation. A similar behavior was observed in the depressive behavior of rats. It was observed that the decrease in withdrawal response was R2 (17.25 s) > R1 (13.8 s) > R4 (12.27 s) > R3 (9.75 s) > R5 (8.9 s). The *p* value was found to be in the following order: R2 (*p* < 0.0001), >R1 (*p* < 0.041), >R4 (*p* < 0.001), and R3 (ns) in comparison to control R5. The significance level was R2 (****), >R1 (***), >R4 (**), and R3 (ns). These findings confirm that co-combination gel successfully reduces neuropathic pain in the animal model as depicted by their behavior and is represented by graphical presentation.

## Figures and Tables

**Figure 1 molecules-27-04433-f001:**
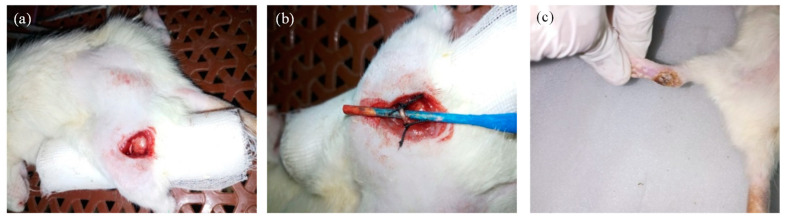
(**a**) Exposing Sciatic nerve, (**b**) Constriction of Sciatic nerve, and (**c**) Scarring of the skin at heel.

**Figure 2 molecules-27-04433-f002:**
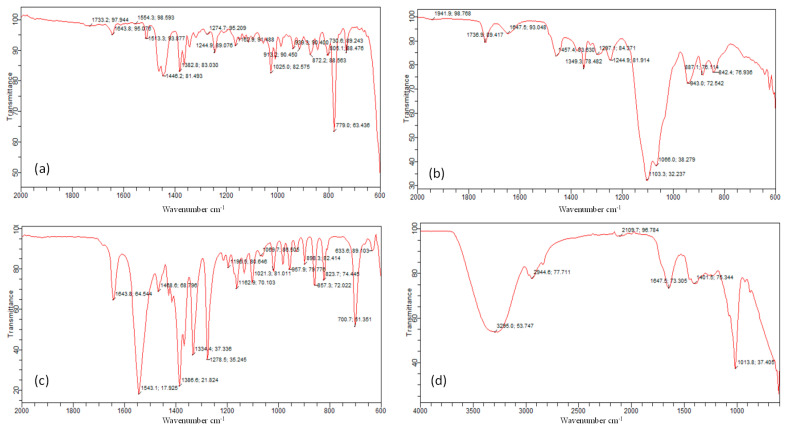
The compatibility studies: (i) excipients;(**a**) Frankincense oil pure, (**b**) Smix, (**c**) Pregabalin pure, (**d**) *Withania coagulans* extract pure ((ii) topical gels; (**e**) Extract loaded gel, (**f**) co-combination gel, (**g**) drug loaded gel).

**Figure 3 molecules-27-04433-f003:**
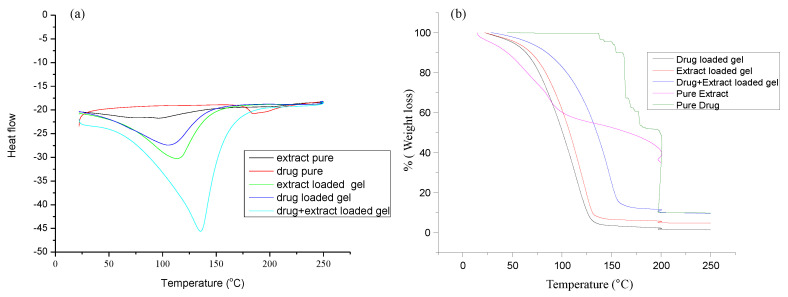
DSC (**a**) and TGA (**b**) curves of the active ingredients and prepared formulations.

**Figure 4 molecules-27-04433-f004:**
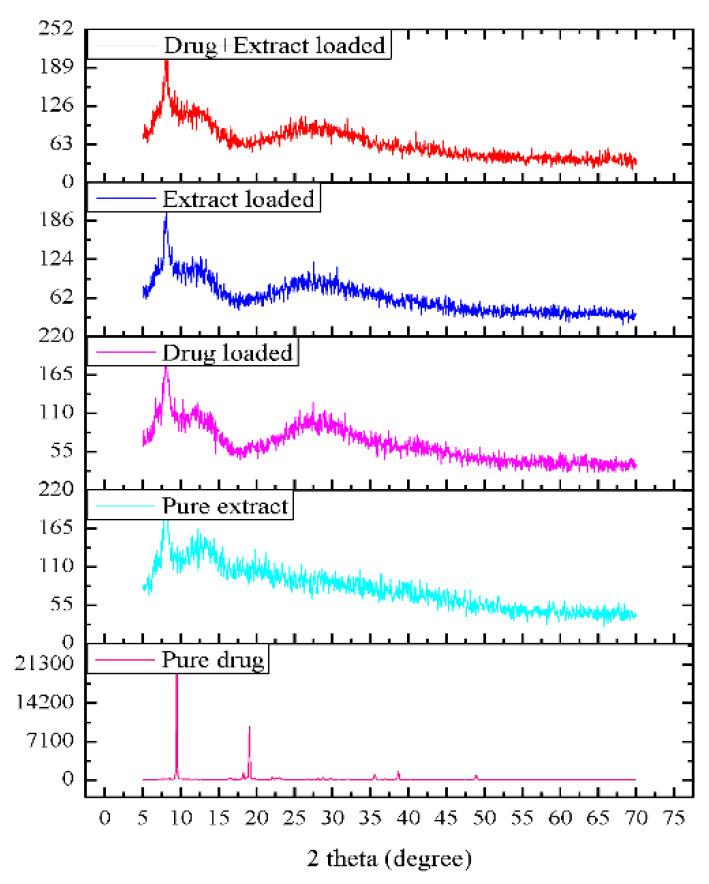
PXRD pattern of active ingredients and the topical gels.

**Figure 5 molecules-27-04433-f005:**
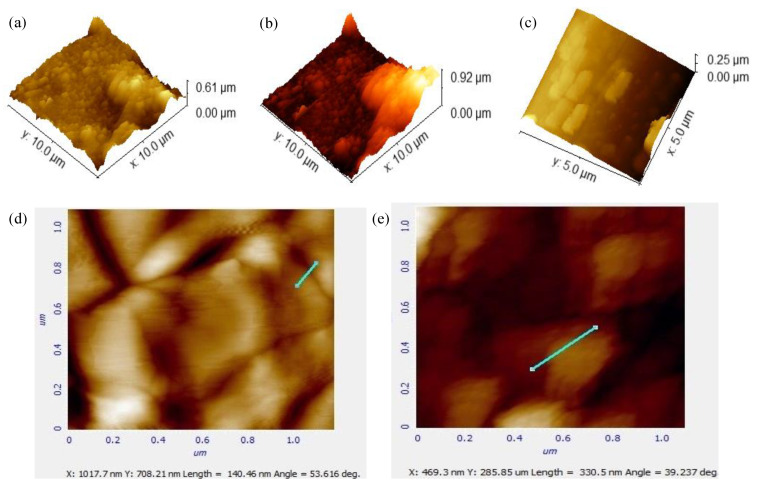
AFM images of (**a**) pure Pregabalin, (**b**) Pregabalin-loaded gel, (**c**) Pure extract (**d**), Pregabalin + extract-loaded gel, and (**e**) Extract-loaded gel.

**Figure 6 molecules-27-04433-f006:**
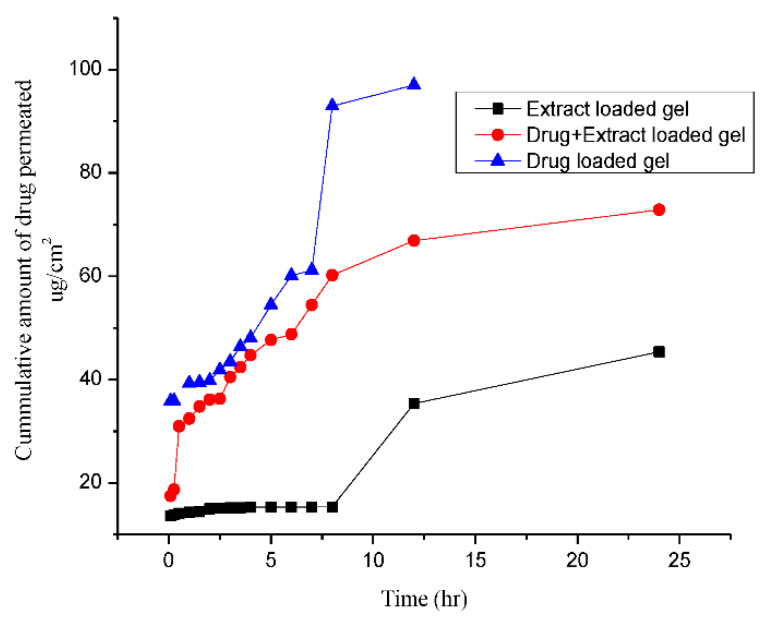
Drug permeability studies from different formulations.

**Figure 7 molecules-27-04433-f007:**
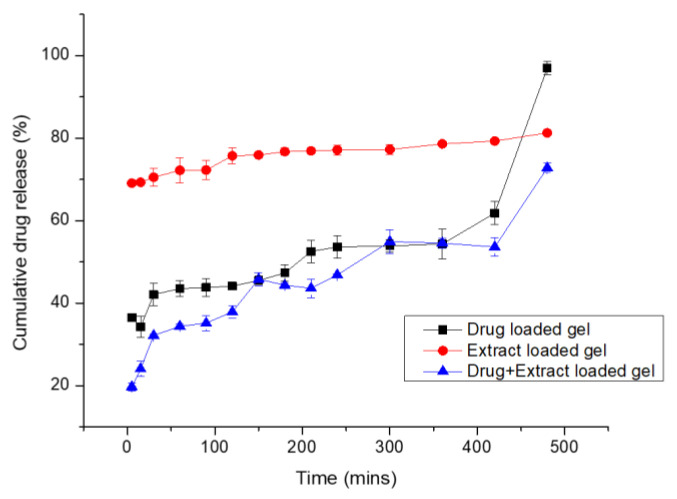
Graph between time (mins) and % release of different gel formulations.

**Figure 8 molecules-27-04433-f008:**
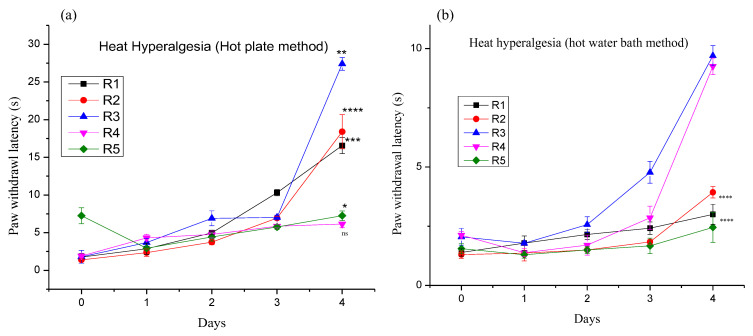
Heat hyperalgesia: (**a**) Hot plate and (**b**) hot water bath method. Where, *, **, ***, ****, ^ns^ represents very less significant, less significant, more significant, very much significant and non significant.

**Figure 9 molecules-27-04433-f009:**
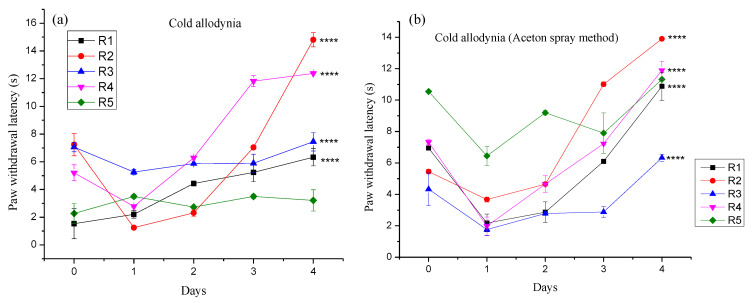
Cold allodynia (**a**) and acetone spray method (**b**). Where, **** represents very much significant.

**Figure 10 molecules-27-04433-f010:**
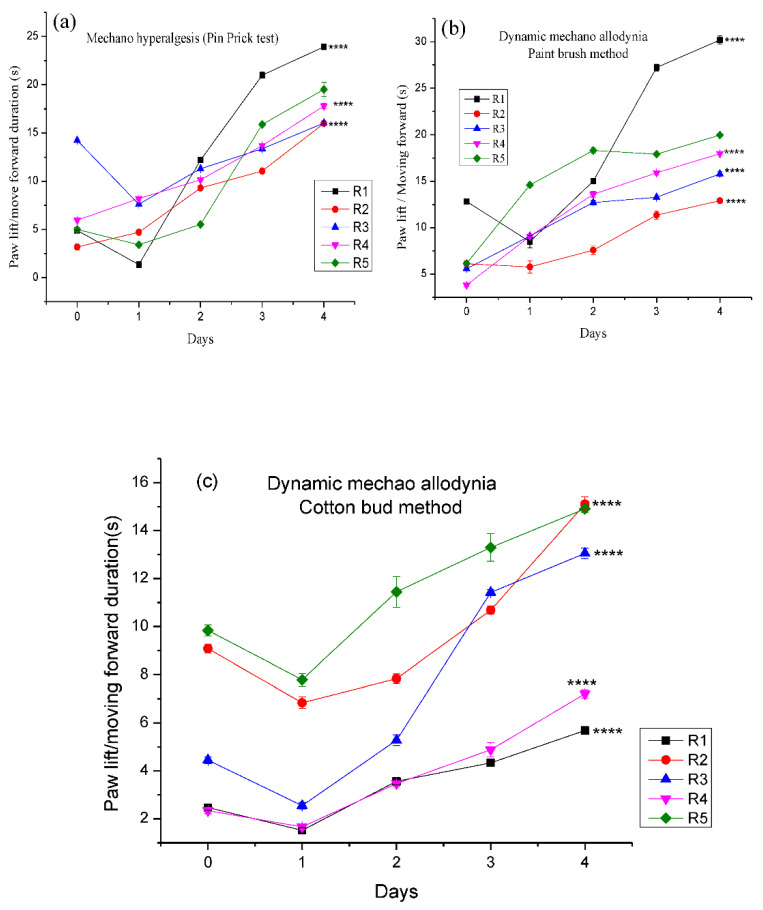
Mechano-hyperalgesia and mechano-allodynia: (**a**) pin prick test, (**b**) paint brush method, and (**c**) cotton bud method. Where, **** represents very much significant.

**Figure 11 molecules-27-04433-f011:**
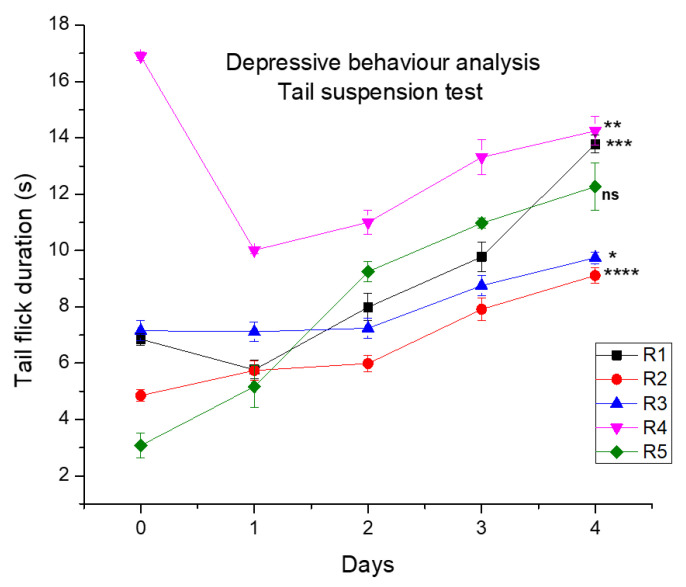
Tail flick test for the analysis of depression behavior gels. Where, *, **, ***, **** and ^ns^ represent, less significant, more significant, very much significant and non significant.

**Table 1 molecules-27-04433-t001:** Peaks showing bending stretching of different excipients and topical gels.

Ingredients	N-H Bend	N-O Stretch	C-H Bend	C-O Stretch	O-H Bend	CH_2_-CH_3_
Pregabalin	1643 cm^−1^	1543 cm^−1^	1468 cm^−1^	1279 cm^−1^	857–933 cm^−1^ [33].	633 cm^−1^
Extract				1013 cm^−1^		2944 cm^−1^
Frankincense oil	1643 cm^−1^	1554 cm^−1^	1446 cm^−1^	1274 cm^−1^	939 cm^−1^	
Smix	1647 cm^−1^		1457 cm^−1^	1297 cm^−1^	943 cm^−1^	
Drug loaded gel	1640 cm^−1^		1457 cm^−1^			618 cm^−1^
Extract loaded gel	1636 cm^−1^		1457 cm^−1^			
Drug + extract loaded gel	1640 cm^−1^		1461 cm^−1^			622 cm^−1^

**Table 2 molecules-27-04433-t002:** Kinetic modeling showing mechanism of drug release from topical gels.

Sr. no	Zero Order		First Order		Krosmeyer Pappas			Higuchi		Hixson Crowell		Result
	K_o_	R^2^	K_1_	R^2^	KKP	R^2^	*n*	KH	R^2^	KHC	R^2^	
E	0.256	0.909	0.124	0.627	0.124	0.627	0.124	5.035	0.952	0.003	0.9325	Higuchi + fickian
D	0.199	0.862	0.004	0.783	18.519	0.742	0.206	3.68	0.802	0.001	0.8049	Zero order + non fickian
E + D	0.173	0.947	0.003	0.950	10.688	0.947	0.282	3.24	0.958	0.001	0.001	Higuchi + fickian

## Data Availability

Most of the data has been published in main article. Raw data to reproduce these findings cannot be shared at this time due to technical and time limitations.

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
