# Peer review of "Co-Combination of Pregabalin and Withania coagulans-Extract-Loaded Topical Gel Alleviates Allodynia and Hyperalgesia in the Chronic Sciatic Nerve Constriction Injury for Neuropathic Pain in Animal Model"

_molecules, 2022, doi:10.3390/molecules27144433_

Round 1
Reviewer 1 Report
The manuscript entitled “Co-combination of Pregabalin and Withania coagulans extract loaded topical gel alleviates allodynia and hyperalgesia in the chronic sciatic nerve constriction injury for neuropathic pain in animal model” have described the fabrication of co- combination gel using Pregabalin and Withania coagulans fruit extract to validates its effectiveness for neuropathic pain in chronic constriction injury (CCI) rat models. Authors should correct manuscript according to the suggestion:
Minor issues:
Line 62-65: Authors should give information which phytochemicals are responsible for health benefits of Withania
Line 85: why Authors using chloroform and methanol for extract preparation? please explain
Line 149 and 249 it should be "Withania" Line 403 it should be 0.001
Figure 2: please give a Figure in better quality
References should be according to Molecules authors guide
e.g. in all references all authors should be given: Author 1, A.B.; Author 2, C.D. Title of the article. Abbreviated Journal Name Year, Volume, page range; pleas carful correct references
In Conclusions some results should be given
Author Response
Answer to Reviewer 1 queries

Reviewer 2 Report
The manuscript reports the physicochemical and pharmacological characterization of a Pregabalin-Withania coagulans-associated topic formulation for neuropathic pain. Both theme and and results are relevant for science and fits with scope of Molecules. However, the manuscript could not be accepted in the present form. Therefore, some deep corrections must be performed prior to acceptance:
- Even being a well-known and reported plant species, authors must ensure they have worked with the correct species. Therefore, a botanical assessment is necessary in order to elucidate this point.
- In 2.2.1 subtopic, "obtention" is rather preferable then "isolation". Furthermore, the reference for formulation is not on reference list (Agshar et al., 2021). It is important to cite and allow us evaluate the novelty of this work.
- 2.2.2 subtopic is named "2.2.1". Please correct.
- In 2.4 topic, "OCED" must be "OECD". The reference for this alternative test must me adequately described in reference list, in order to allow knowledge of methods and reproductibility.
- In 2.5 topic, line 149: "Withania".
- In some subtopics from 2.7, there are some procedures without cited proper references. Please correct.
- The Results are not discussed at all! Authors must strongly use previous reports from scientific literature in order to express relevance and novelty for those findings. As presented, the manuscript could not be accepted.
- The Conclusion is very poor and need to bring some lacking outcomes regarding physicochemical stability, and how valuable is the combination of pregabalin-extract rather than both ones in separate.
Author Response
Answer to Reviewer 2 queries

Round 2
Reviewer 2 Report
The manuscript was adequately revised after suggestions.